# Structural basis of THC analog activity at the Cannabinoid 1 receptor

Thor S. Thorsen [1,8,10], Yashraj Kulkarni [1,10], David A. Sykes [2,3], Andreas Bøggild [4], Taner Drace[4], Pattarin Hompluem[2,3], Christos Iliopoulos-Tsoutsouvas[5], Spyros P. Nikas [5], Henrik Daver [1,9], Alexandros Makriyannis[5,6], Poul Nissen[4,7], Michael Gajhede [1], Dmitry B. Veprintsev [2,3,11], Thomas Boesen[4,7,11], Jette S. Kastrup[1,11] & David E. Gloriam [1,11] ✉

Tetrahydrocannabinol (THC) is the principal psychoactive compound derived from the cannabis plant *Cannabis sativa* and approved for emetic conditions, appetite stimulation and sleep apnea relief. THC's psychoactive actions are mediated primarily by the cannabinoid receptor $CB_1$. Here, we determine the cryo-EM structure of HU210, a THC analog and widely used tool compound, bound to $CB_1$ and its primary transducer, $G_{i1}$. We leverage this structure for docking and 1000 ns molecular dynamics simulations of THC and 10 structural analogs delineating their spatiotemporal interactions at the molecular level. Furthermore, we pharmacologically profile their recruitment of $G_i$ and β-arrestins and reversibility of binding from an active complex. By combining detailed $CB_1$ structural information with molecular models and signaling data we uncover the differential spatiotemporal interactions these ligands make to receptors governing potency, efficacy, bias and kinetics. This may help explain the actions of abused substances, advance fundamental receptor activation studies and design better medicines.

$Δ^9$-tetrahydrocannabinol (THC) is the principal psychoactive compound derived from the cannabis plant, Cannabis sativa. The beneficial physiological effects of THC include analgesia and appetite stimulation[1], while harmful effects include e.g., paranoia and anxiety[2]. As a marketed pharmaceutical Dronabinol, THC is approved as an antiemetic, appetite stimulant, and reliever of sleep apnea[3]. The THC analog Cannabidiol is used for managing anxiety, insomnia, pediatric epilepsy, and chronic pain, and these two drugs are combined in Nabiximols, treating neuropathic pain and spasticity in multiple sclerosis. A synthetic THC analog, Nabilone, is also available as an antiemetic and adjunct analgesic for neuropathic pain effective in relieving fibromyalgia[4] and multiple sclerosis[5]. Clinical trials are investigating cannabinoids for treating glioblastoma and a variety of neurological disorders, including Parkinson's disease, Alzheimer's disease, Huntington's disease, amyotrophic lateral sclerosis, traumatic brain injury, and stroke[6].

The main targets of THC are the cannabinoid receptors, $CB_1$ and $CB_2$. Psychoactive effects are mediated by $CB_1$ in the cerebral cortex,

[1]Department of Drug Design and Pharmacology, University of Copenhagen, Copenhagen, Denmark. [2]Centre of Membrane Proteins and Receptors (COMPARE), University of Nottingham, Nottingham, Midlands, UK. [3]Division of Physiology, Pharmacology & Neuroscience, School of Life Sciences, University of Nottingham, Nottingham, UK. [4]Interdisciplinary Nanoscience Center and Department of Molecular Biology & Genetics, Aarhus University, Aarhus, Denmark. [5]Center for Drug Discovery and Department of Pharmaceutical Sciences, Northeastern University, Boston, MA, US. [6]Center for Drug Discovery and Department of Chemistry and Chemical Biology, Northeastern University, Boston, MA, US. [7]Danish Research Institute of Translational Neuroscience – DANDRITE, Nordic EMBL Partnership for Molecular Medicine, Aarhus University, Denmark, Aarhus, Denmark. [8]Present address: Nordic Virtual Pastures, BioInnovation Institute, København N, Denmark. [9]Present address: H. Lundbeck A/S, Valby, Denmark. [10]These authors contributed equally: Thor S. Thorsen, Yashraj Kulkarni. [11]These authors jointly supervised this work: Dmitry B. Veprintsev, Thomas Boesen, Jette S. Kastrup, David E. Gloriam. ✉e-mail: david.gloriam@sund.ku.dk

cerebellum, and basal ganglia, whereas $CB_2$ is mainly expressed in cells of the immune system[7]. $CB_1$ mediates its signaling primarily by $G_{i/o}$ that inhibits adenylate cyclase which in turn decreases the cellular concentration of cyclic AMP[8]. In addition, $CB_1$ also couples to $G_s$, $G_{q/11}$, and arrestins[9–11]. Several $CB_1$ ligands exhibit pathway-biased signaling[12,13] i.e., preferential activation of a specific transducer pathway. Biased signaling can lead to functionally selective responses, paving the way for the design of drugs with improved therapeutic profiles and fewer side effects[14,15]. The multiple pathways, biased ligands, and clinical indications make $CB_1$ an appealing system for studying biased signaling.

The structural coverage of complexes of THC analogs and cannabinoid receptors presently spans AM841[16,17], CP55940[18–20] and AM11542[16] bound to $CB_1$, and AM12033[17], CP55940[21] and HU243[21] bound to $CB_2$. The resolution of the structures spans 2.8–3.4 Å and three structures are fully activated cryo-EM complexes with $G_{i1-2}$, whereas the AM11542-$CB_1$ complex is a crystal structure with a fusion protein, flavodoxin, that restrains TM6 in an active state. The overall coverage of the CB1 protein sequence in active-state structures ranges between 58% and 62% of residues (Supplementary Table 1). This limited coverage is due to the intrinsic flexibility of the N-/C-termini and the long third intracellular loop (ICL3).

THC has a tricyclic core consisting of phenyl, pyran, and cyclohexene rings that are indexed A, B, and C, respectively (Supplementary Fig. 1). Opening of the pyran ring leads to compounds referred to as Cannabidiols that have weaker affinity to $CB_{1-2}$ and psychoactivity[22] than the THC class. Numerous THC analogs have been developed and published, among which HU-210 is one of the most potent tool compounds. HU210 is 100–800 times more potent than naturally sourced THC in discrimination tests in animal studies[23] and has a binding affinity at human $CB_1$ that is ~700 higher than for THC[24]. HU243 is a close structural analog of HU210, differing just by the lack of a double bond in C8-C9 of the C ring, sharing its high potency[24]. Additional potent synthetic THC analogs include the widely used tool compound, CP55940 as well as AM841 and AM11542. In contrast, the phytocannabinoid Cannabinol[25] and synthetic ligands L759633[26] and JWH133[27] have lower affinities and potencies at $CB_1$ and less strong psychoactive effects than THC. Inverse agonism is observed for Tetrahydrocannabivarin (THCv) which has the opposite effect of THC by suppressing appetite and controlling blood glucose levels[28], making it a potential treatment for obesity and diabetes.

Structure-activity relationship studies of THC have focused mainly on the three major pharmacophore features: (i) the phenolic hydroxyl group at the C1 position on the A-ring, (ii) the methyl group at the C9 position on the C-ring, and (iii) the alkyl side chain at the C3 position on the A-ring[29–31] (Supplementary Fig. 1). C1 phenolic hydroxyl removal (JWH133) and etherification (L759633) increase selectivity for $CB_2$ over $CB_1$ activation by 153-fold[27] and 200-fold[26], respectively. Substitution of the C9 methyl with hydroxyl (CP55940) or carbonyl (Nabilone) groups improves affinity, but not selectivity[31]. A shorter and longer alkyl tail compared to THC decreases and increases binding affinity and potency, respectively[32–34], and five to eight carbons are considered optimal[32,35]. Elongation with large halogens at the terminal position (as in AM11542) increases binding affinity. Binding affinity can also be enhanced by methyl substitution at the C1' and C2' positions[31]. Here, 1',2'-substitution gives as good binding affinity but 1',1'-dimethyl substituted compounds are easier to synthesize. Hitherto, these and other structure-activity studies have predominantly focused on analog substitutions correlated to binding activity or selectivity, and assessment of potency has been typically addressed as effective dose in vivo[29–31]. This leaves a need for more molecular mechanistic studies covering ligand-receptor interactions and dissecting additional pharmacological parameters under consistent experimental conditions.

Here, we benchmark the potency and efficacy of $\Delta^9$-THC and 10 analogs (Supplementary Table 2) for the recruitment of the primary transducer family, $G_{i/o}$ and β-arrestin. Furthermore, we solve the cryo-EM structure of the HU210/$CB_1$/$G_{i1}$ signaling complex and leverage this template to obtain consistent ligand binding modes and spatio-temporal interactions through 1000 ns molecular dynamics (MD) simulations. Combining these approaches, we expand the structural rationale of THC analogs beyond ligand moieties to dynamic receptor interactions delineating determinants for potency, efficacy, kinetics, and biased signaling.

The ligands included in this study include $\Delta^9$-THC and analogs of both $\Delta^9$-THC and $\Delta^8$-THC. $\Delta^8$-THC is a structural isomer of $\Delta^9$-THC, differing only in the location of the double bond in the C-ring. Whilst $\Delta^9$-THC has a double bond at the C9 position with C10 (Supplementary Fig. 1), $\Delta^8$-THC has a double bond at the C8 position with C9. Hexahydrocannabinol (HHC) is a derivative of $\Delta^9$-THC that lacks a double bond in the C-ring. Therefore, the analogs included in this work can be classified in the following manner. $\Delta^9$-THC analog: Tetrahydrocannabivarin (THCv); $\Delta^8$-THC analogs: AM11542, HU210, JWH133, L759633; HHC analogs: AM841, CP55940, HU243, Nabilone. Cannabinol differs from THC by having an aromatic C-ring.

## Results

### Cryo-EM structure of the HU210/$CB_1$/$G_{i1}$ signaling complex

The human $CB_1$ receptor (residues 2-472) and a $G_{i1}$ double mutant (G203A, A326S) with increased propensity to form complex with the receptor[36,37] were co-expressed in Sf9 insect cells. A stable HU210/$CB_1$/$G_{i1}$ complex was formed in cell membranes by binding of the ScFv16 fragment antibody (Fab), and hydrolysis of released GDP nucleotides using Apyrase enzyme. The complex was purified in LMNG detergent micelles, using a M1 anti-Flag affinity resin and size exclusion chromatography. The complex was blotted onto carbon gold grids and imaged using a Titan Krios microscope (Supplementary Fig. 2). The derived HU210/$CB_1$/$G_{i1}$ structure (Fig. 1a and Supplementary Fig. 3) has a global resolution of 2.9 Å and large overall similarity to published $CB_1$/$G_{i1-2}$ complexes (Fig. 1b and Supplementary Table 3). The HU210-bound structure has the largest coverage of receptor residues among all reported active state $CB_1$ structures[16–19,38,39] (Supplementary Table 1). It covers 294 residues, which is four residue backbones and seven sidechains more than the second most complete structure[19]. The structure reveals more of the N-terminus wherein residues 100–105 form a β-hairpin-like bend with van der Waals contacts to residues F108[N-ter], Q261[ECL2], C264[ECL2], and M371[ECL3] (Fig. 1c). Intriguingly, we also observed structural changes in the G protein binding site, far away from the ligand binding site. The HU210 complex (with $G_{i1}$) and CP55940 complex[19] (with PAM and $G_{i2}$) structures demonstrate distinct conformations of the modeled residues in the intracellular loop ICL3 (Fig. 1d). Altogether, the HU210-bound structure expands our insights into the $CB_1$/$G_{i1}$ signaling complex and adds a high-resolution template for THC structure-activity studies. HU210 shares binding site with the THC analogs AM841[16,17], CP55940[18,19] (cryo-EM $G_{i1-2}$ complexes), AM11542[16] (crystal structure with a fusion protein, flavodoxin in ICL3) (Fig. 1e, f) bound to $CB_1$ and AM12033-bound $CB_2$[17] ($G_{i1}$ cryo-EM structure). Conformational clustering of the transmembrane helices of all active-state $CB_1$ structures shows that ligands have a large impact on the structural similarity of receptors (Fig. 1g). Whereas the receptors first separate into crystal structures with fusion proteins (inducing an active-like conformation) and cryo-EM structures with G proteins, HU210-bound $CB_1$ then groups closest to the cryo-EM complex with the closest analog of HU210, AM841 (Supplementary Table 1). These two ligands have a Tanimoto coefficient of 0.82 and their respective $CB_1$/$G_{i1}$ structure complexes have an RMSD value of 0.81 Å. Furthermore, the crystal and cryo-EM structures grouping the closest contain the same ligand, AM841 (Fig. 1g). This demonstrates a notably high conformational impact of the relatively small ligand on the overall $CB_1$/$G_i$ structure. Taken together, we extend the number of THC analogs with a $CB_{1-2}$ structure to five and find that these bind with similar

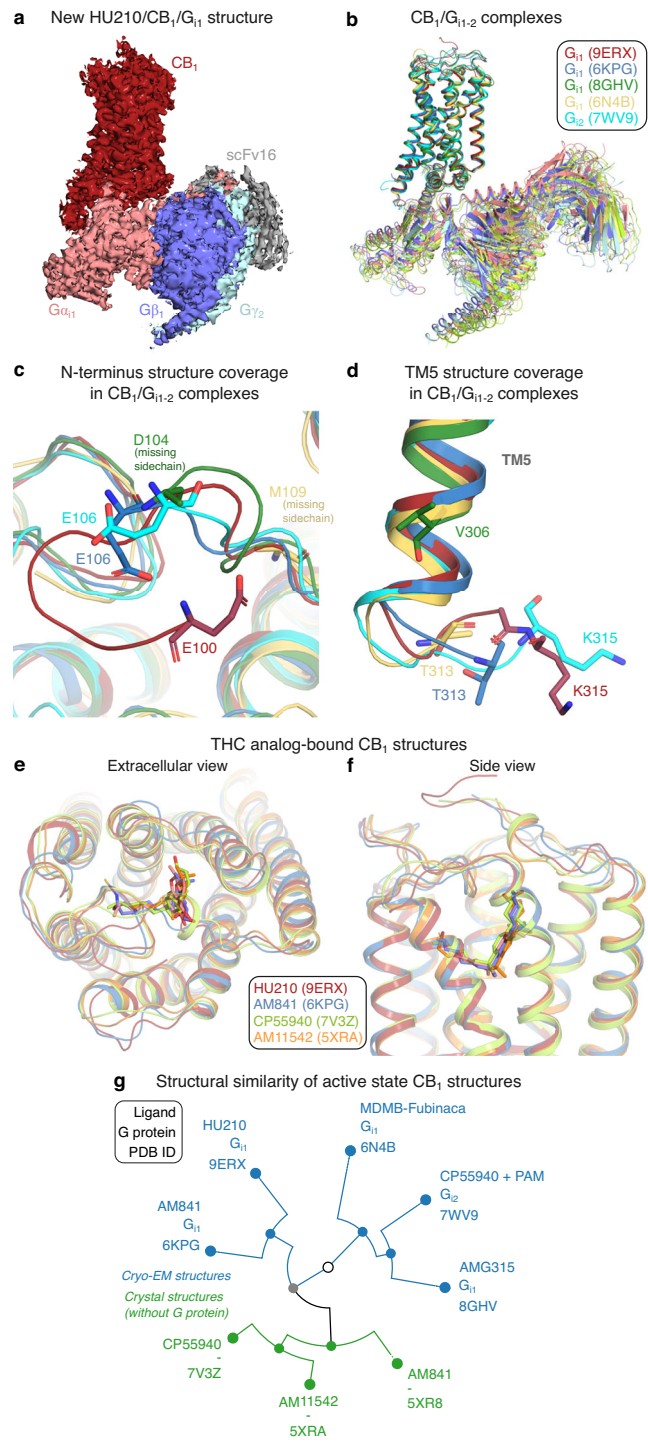

**Fig. 1 | Active state CB₁ structures – completeness and complexes with Gᵢ and THC analogs. a** Cryo-EM density map (sharpened) of the HU210/CB₁/Gᵢ₁ signaling complex. **b** CB₁-Gᵢ complexes[17,19,38,39] superposed based on the seven transmembrane helix backbone atoms. N-terminal (**c**) and TM5 (**d**) segments uniquely covered by the HU210/CB₁/Gᵢ₁ structure. The shown residues are the terminal residue in each structure. Extracellular (**e**) and (**f**) side views of THC analog/CB₁ structure complexes. **g** Structure similarity tree of active state CB₁ structures based on a conformational clustering in GPCRdb making a superposition-independent all-to-all residue distance comparisons across the transmembrane domain[51].

conformations in the same site and have a clear impact on the receptor conformation. These structures provide a strong support for structure-activity relationship analysis aiming to delineate common and unique receptor interactions (below).

## THC analogs exhibit diverse pharmacological activity

We selected 10 structurally diverse analogs of THC for investigation by docking, molecular dynamics, and pharmacological assaying (Supplementary Fig. 1). This includes C1 phenolic hydroxyl removal (JWH133) and etherification (L759633), and C9 methyl replacement with carbonyl (Nabilone), hydroxy (CP55940) or hydroxymethyl (AM841, HU210, and HU243) groups. For the C3 alkyl tail, the analogs span propyl (THCv), butyl (JWH133), pentyl (THC and Cannabinol), heptyl (L759633, Nabilone, CP55940, HU243, and HU210) as well as heptyl substituted with isothiocyanate (AM841) or Bromo (AM11542) groups. Of these 11 ligands, eight have an initial branching (1′,1′-dimethyl substitution) of the alkyl chain whereas THC, THCv, and Cannabinol lack this feature. Altogether, the selected 11 ligands allow for informative and efficient characterization of structure-function covering the major determinants[29–31].

We profiled the recruitment of the primary transducer, Gᵢ/ₒ to CB₁ for THC, the 10 analogs and the endocannabinoids 2-Arachidonoylglycerol (2AG) and N-arachidonoylethanolamine (AEA, Fig. 2 and Supplementary Fig. 1). We find that the tested ligands fall into five major groups by their activity: (i) low-potency/high-efficacy (2AG and AEA), (ii) high-potency/high-efficacy (AM841, AM11542, CP55940, HU210, HU243 and Nabilone), (iii) low-potency/low-efficacy (Cannabinol, JWH133, L759633), (iv) high-potency/low-efficacy (THC) and (v) inverse agonist/antagonist (THCv) (Fig. 3). The endocannabinoid 2AG has the highest efficacy and was selected as the reference ligand for comparisons of efficacy (E$_{max}$ = 100%).

## SAR of high-potency/high-efficacy THC analogs

All high-potency/high-efficacy THC analogs (AM841, AM11542, CP55940, HU210, HU243, and Nabilone) have an alkyl tail consisting of seven carbons—two more than the reference ligand THC. Through molecular dynamics (MD) simulations, we find that the high activity of the long alkyl chains may be attributed to favorable contacts to T197$^{3x33}$, I271$^{ECL2}$, Y275$^{5x39}$, and W279$^{5x43}$ that are 21, 26, 49, and 20% more frequent, respectively relative to THC (Fig. 4c). Conversely, a comparison with AM841, shows that a further extension of HU243 with an isothiocyanate moiety lowers ligand efficacy from 74% to 61% showing that a too long, or polar alkyl chain is not favorable. In MD simulations, AM841 positions the isothiocyanate substituent in an angled conformation between TM3 and TM4 where it induces a rotamer shift of Y275$^{5x39}$ indicating a strained and/or higher-energy binding conformation (Fig. 5a). Like AM841, AM11542 also has a further elongated alkyl tail, being bromo-substituted, and its efficacy is lower than that of AM841.

The low efficacy of AM11542, compared to other analogs in this group, could also be attributed to the lack of a hydrogen bonding substituent at C9 at the opposite end of the ligand. Furthermore, Nabilone, which has the lowest potency in this group, has a carbonyl at C9 indicating that hydrogen bond donor capacity is essential for high activity. Consistently, the three agonists with the highest efficacy and potency have a hydroxyl (CP55940) or hydroxymethyl (AM841, HU210, and HU243) that can be hydrogen bond donors. In our MD simulations, CP55940's C9 hydroxyl forms a very stable hydrogen bond to the backbone carbonyl of I267$^{ECL2}$ (Fig. 5b). The hydroxymethyl-containing ligands share the hydrogen bond with I267$^{ECL2}$ and have additional van der Waals contacts with H178$^{2x65}$. Another difference in the C9 carbon is the hybridization state, which is sp³ for AM841, CP55940, and HU243, and sp² for AM11542, HU210, and Nabilone. Our MD simulations show that ligands with sp³-hybridized C9 carbons, especially CP55940 and HU243, have a more stable hydrogen bond to the backbone carbonyl of I267$^{ECL2}$ than observed for sp²-hybridized agonists (Fig. 5f and Supplementary Fig. 4c, d). Nabilone instead has a unique, although low frequent, hydrogen bond between its C9 carbonyl and H178$^{2x65}$.

All the ligands in this group have a 1′,1′-dimethyl substitution, which is important for activity. Uniquely, CP55940 lacks the B-ring and

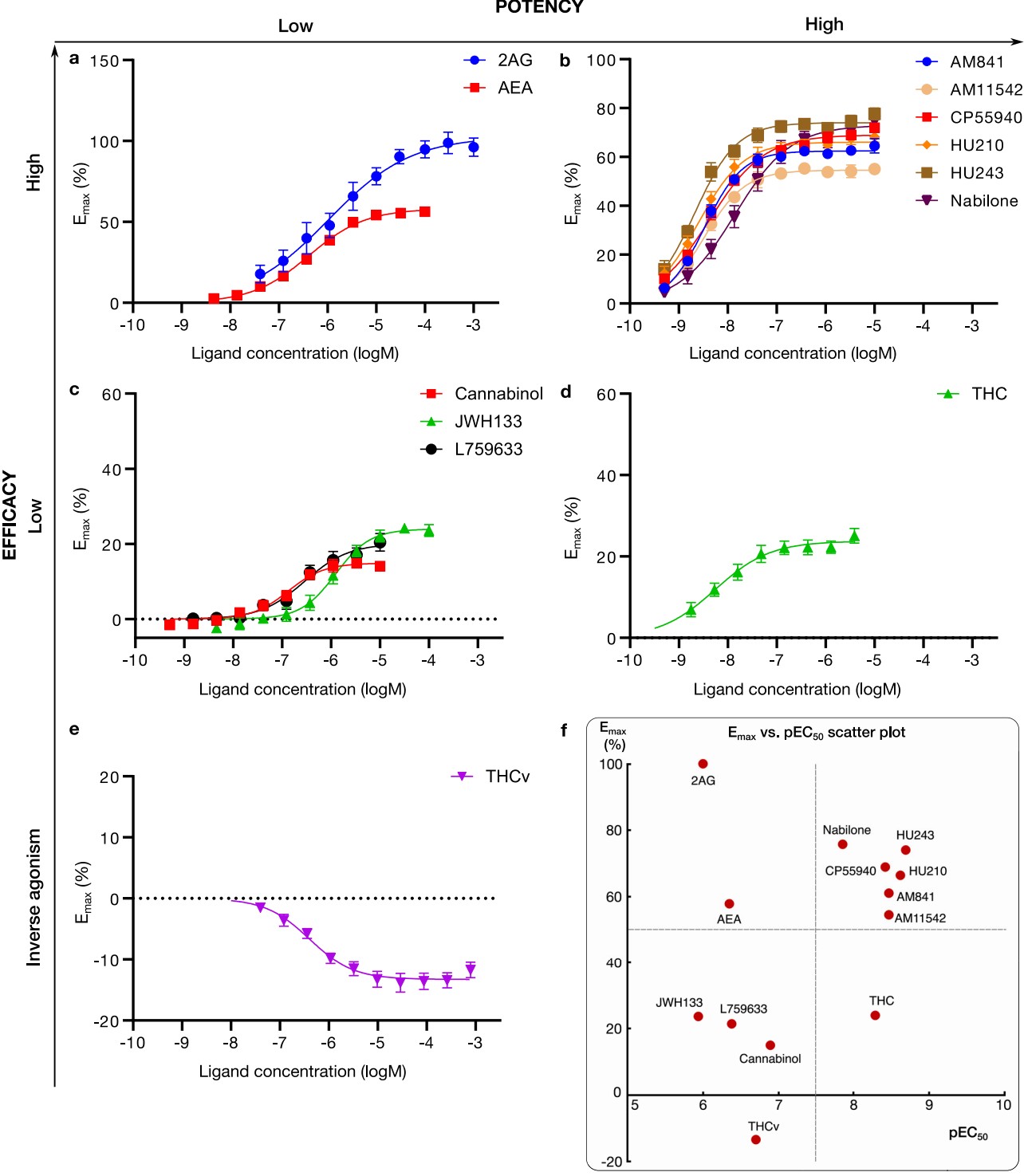

**Fig. 2 | G protein recruitment of THC analogs and endogenous agonists via CB₁.** Ligand concentration-response curves for (**a**) low-potency/high-efficacy agonists, (**b**) high-potency/high-efficacy agonists, (**c**) low-potency/low-efficacy agonists, (**d**) high-potency/low-efficacy agonist, and (**e**) inverse agonist, from mG$_i$ recruitment. **f** Scatter plot of ligand pEC$_{50}$ (x-axis) and E$_{max}$ (y-axis) in the mG$_i$ experiments. HEK293-TR cells stably co-expressing a combination of either the CB₁-NlucC and NES-venus-mGsi or NES-venus-β-arrestin were used. Recruitment data, expressed as a % of the maximal response produced by the endogenous cannabinoid 2AG, is plotted versus log concentration for the indicated ligands. Data were fitted to a four-parameter logistic equation; mean ± S.E.M. of at least 3 independent experiments. N-values for the individual ligand CRCs are: 2AG (6), AEA (9), AM841 (6), AM11542 (6), CP55940 (17), HU210 (6), HU243 (6), Nabilone (6), Cannabinol (6), JWH133 (5), L759633 (3), THC (12) and THCv (6).

instead has a rotatable bond between the C10a and C11 atoms. As CP55940 groups close to the middle of the five other analogs it is possible that this flexibility does not affect activity. Alternatively, its effect could be masked by favorable interactions of the unique 6a

hydroxypropyl. MD simulations show that this hydroxypropyl forms a very stable hydrogen bond to S173$^{2x60}$ (Fig. 5b and Supplementary Fig. 4a). Given this observation, and the major increase in conformational freedom at the center of the tricyclic core, the alternative

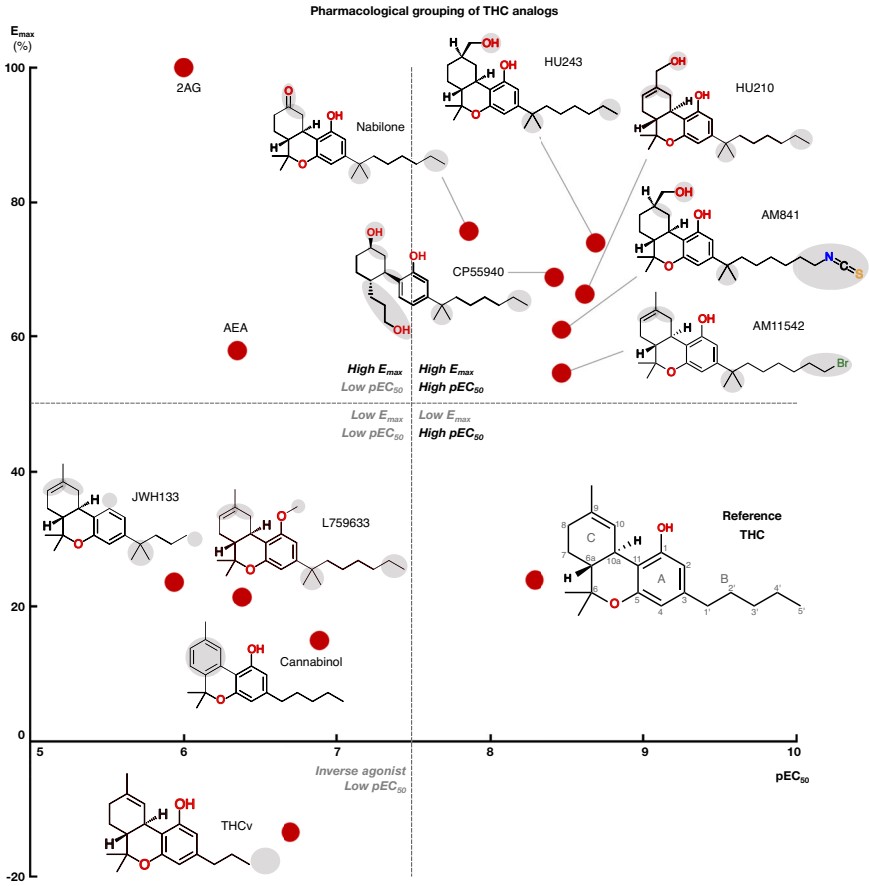

**Fig. 3 | Structure-activity relationships of studied THC analogs.** Scatter plot of ligand pEC$_{50}$ (x-axis) and E$_{max}$ (y-axis) in the mG$_i$ experiments. Dotted lines divide the plot into quadrants of low/high pEC$_{50}$/E$_{max}$. For THC analogs, the 2D structure is shown next to the data points.

explanation of a masked effect seems most plausible. Taken together, this presents a dynamic structural basis for high potency and efficacy that extends and agrees with previous structure-activities studies demonstrating the benefit of the seven-carbon alkyl chain[32,35] and hydrogen bonding functionality (ideally donor) in position C9[31].

## SAR of low-potency/low-efficacy THC analogs
Cannabinol is the single analog that groups closest to THC and it differs only by a single feature – an aromatic benzene in the C-ring. However, this difference alone leads to a 25-fold potency reduction (Supplementary Table 2). JWH133 has the lowest potency of all THC analogs and, like the inverse agonist THCv, has a shorter alkyl tail (one and two carbons fewer, respectively) than THC. JWH133 and L759633 both lack the 1-hydroxyl present in THC and the high-potency/high-efficacy analogs. JWH133, which has no 1-substituent, lacks an electron-donating group on the A-ring—weakening its aromatic interactions—and the hydrogen bond to S383[7x38] in MD (Figs. 4c, 5c and Supplementary Fig. 4b). Furthermore, in L759633, which has a 1-methoxy substituent, this hydrogen bond is 89% less frequent than in THC (Fig. 5d). Notably, although this hydroxyl group is preserved in Cannabinol this ligand also has an infrequent hydrogen bond—due to the conformational change induced by the planar C-ring (Figs. 4b and 5e). Hence, the 1-hydroxyl functionality and S383[7x38] interaction are major contributors to potency. L759633 is structurally similar to HU210 differing only by the 1-substituent (methoxy vs. hydroxyl) and 9-substituent (methyl vs. hydroxymethyl). When considering the structure-activity relationships of all analogs, these two structural differences explain the low potency and efficacy, respectively of L759633.

## SAR of the inverse agonist THCv
Strikingly, THCv only differs from THC by a two-carbon reduction of the alkyl chain but has the opposite modality, inverse agonism, and 39-fold lower potency. The large effect on potency is remarkable, given that the high-potency/high-efficacy analogs also differ by addition of two-carbons relative to THC (albeit addition at a different site) but have similar potencies. The MD simulations show that the short propyl tail of THCv renders a part of the binding pocket vacant. Compared to THC, this leads to a loss of interactions in MD simulations to the residues Y275[5x39], L276[5x40], and I271[ECL2]. However, the MD analysis shows that a more profound characteristic of THCv is a much more dynamic overall binding mode (Fig. 5g and Supplementary Fig. 4e). Throughout the simulations, THCv is anchored by a stable hydrogen bond between the central C1 hydroxyl and S383[7x38] but its tricyclic ring moiety has less stable interactions with F268[ECL2] and L193[3x29] and the two distal ends are considerably more mobile than for THC. Altogether, the inverse agonism of THCv may be attributed in part to lost interactions of the tail but predominantly to the reduced stability of many more interactions across the scaffold.

## G$_{i1}$ recruitment correlates with G$_o$ recruitment and cAMP inhibition
Beyond G$_{i1}$, CB$_1$ couples to other subtypes of the G$_{i/o}$ family (see GproteinDb's *Couplings* page[40]). In the CNS, G$_o$ has been suggested to be a highly abundant G$_{i/o}$ subtype and an investigation has shown that only HU210 induces maximal G$_o$ stimulation, with Δ$^9$-THC and AEA functioning only as partial agonists for the G$_o$ pathway[41]. To explore if the THC analog SAR based on G$_{i1}$ recruitment also reflects G$_o$ recruitment, we correlated maximum efficacy and potency data for eight ligands for

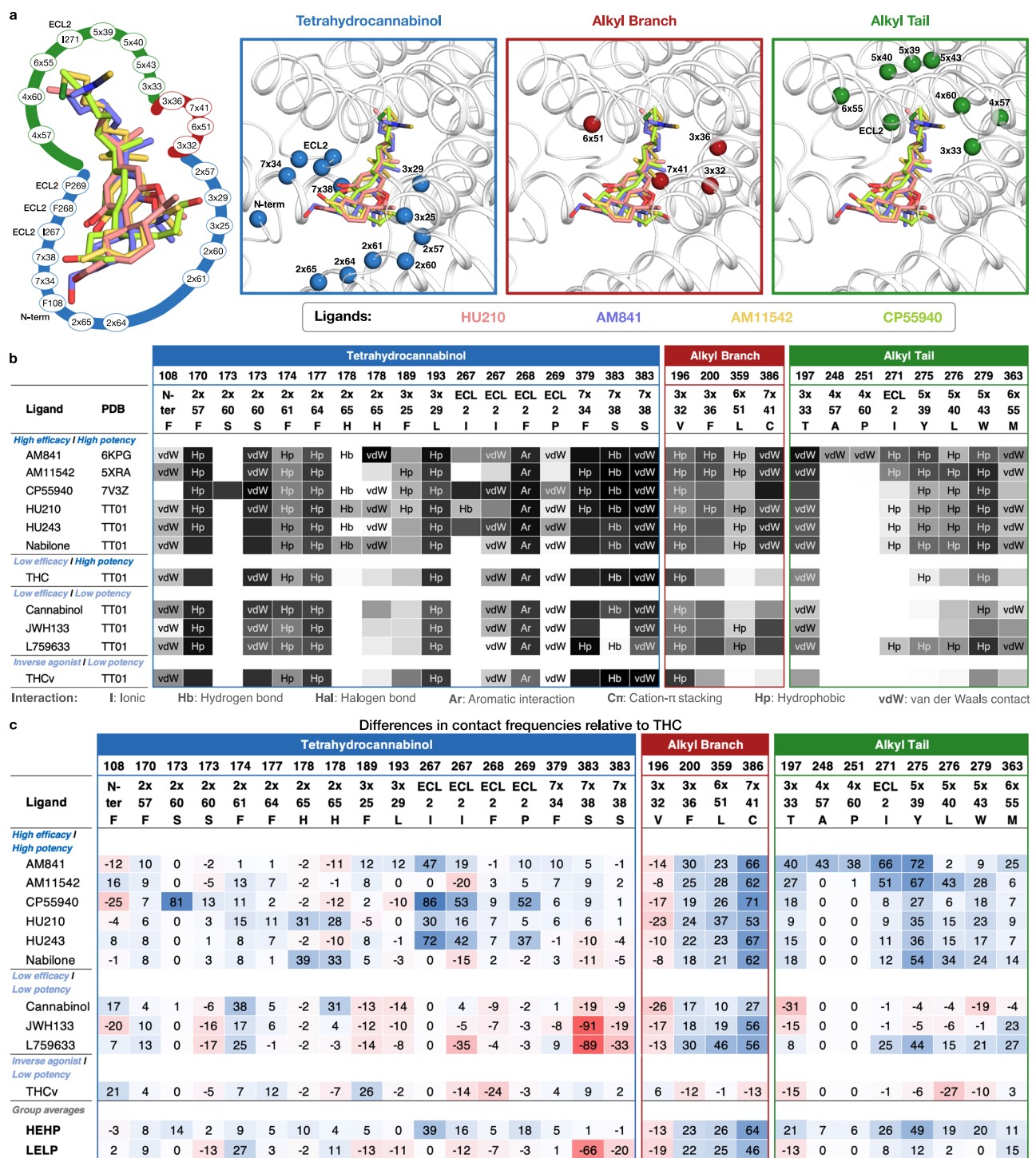

**Fig. 4 | Spatiotemporal THC analog/CB₁ interactions. a** Left: Spatial distribution of the 25 CB₁ residues interacting with HU210 (pink), AM841[17] (blue), AM11542[16] (yellow), and CP55940[18] (green). Right: The three subsites accommodating the tetrahydrocannabinol, alkyl branch, and alkyl tail, respectively. Pocket residues are shown as Cα spheres and labeled with generic residue numbers facilitating comparison of structurally corresponding positions across receptors[78]. **b** Ligand-receptor interaction fingerprints denoting interactions, type, and CB₁ generic residue positions. Interaction labels indicate receptor-ligand contacts in pre-MD starting structure/model (see Methods). Grayscale illustrates frequencies of interactions (%) throughout simulations, averaged across three replicates. Five CB₁ residue positions marked "N-ter" and "ECL2" lack a generic residue number because they are not structurally conserved across receptors. **c** Differences in contact frequencies (in % averaged across three replicates) of THC analogs relative to THC (reference) at each receptor residue. Positive values (blue cells) indicate contact frequencies greater than those in THC, whereas negative values (red cells) indicate contact frequencies lesser than those in THC.

which we previously published GₒA data using similar BRET technology[42] (Supplementary Table 2 and Supplementary Fig. 5). We obtained $E_{max}$ and $pEC_{50}$ $r^2$ values of 0.72 and 0.89, respectively suggesting that the SAR identified for $G_{i1}$ is similarly applicable also to $G_o$ coupling.

Furthermore, G protein recruitment does not necessarily correlate with the strength of cellular signaling outputs. To investigate this, we correlated our $G_{i1}$ recruitment data with literature data on cAMP inhibition for the ligands 2AG, AEA, WIN55212-2, THC, CP55940, and

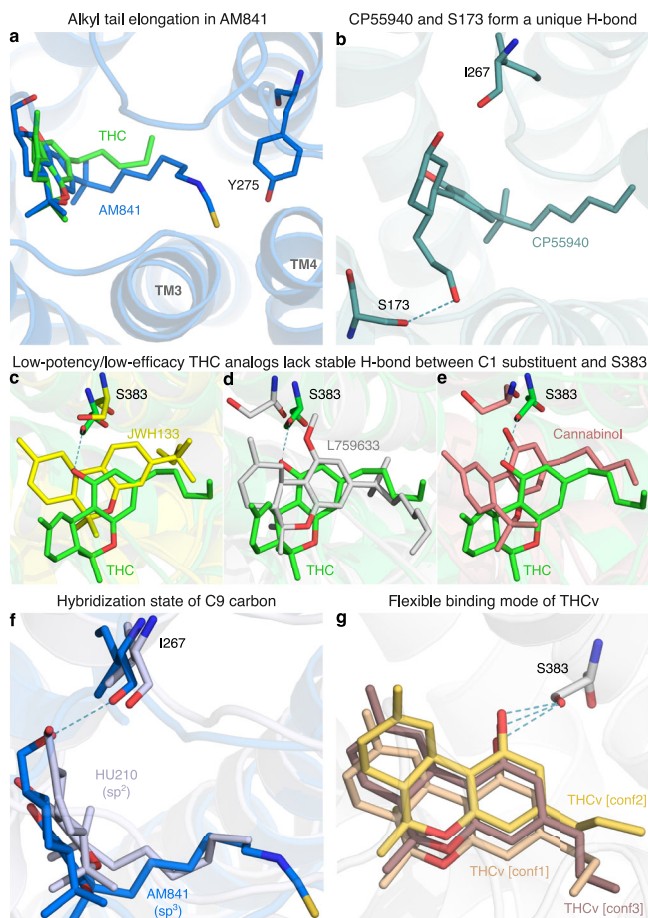

**Fig. 5 | Structural determinants of ligand activity observed in MD simulations.** Visual representations of ligand-receptor interactions from MD simulations that are key determinants of their structure-activity relationship. **a** Conformation of the elongated alkyl tail of AM841 in comparison with THC. **b** Binding mode of CP55940 depicting the unique hydrogen bond interaction between the 6a hydroxypropyl and the sidechain of S173[2x60]. **c–e** Binding modes of low-potency/low-efficacy THC analogs (JWH133, L759633, and Cannabinol, respectively) in comparison with THC. **f** Conformational differences in THC analogs having different hybridization states at C9 carbon of the C-ring; HU210 (sp²) and AM841 (sp³) are shown as representative analogs. **g** Superposition of THCv conformations indicating binding flexibility.

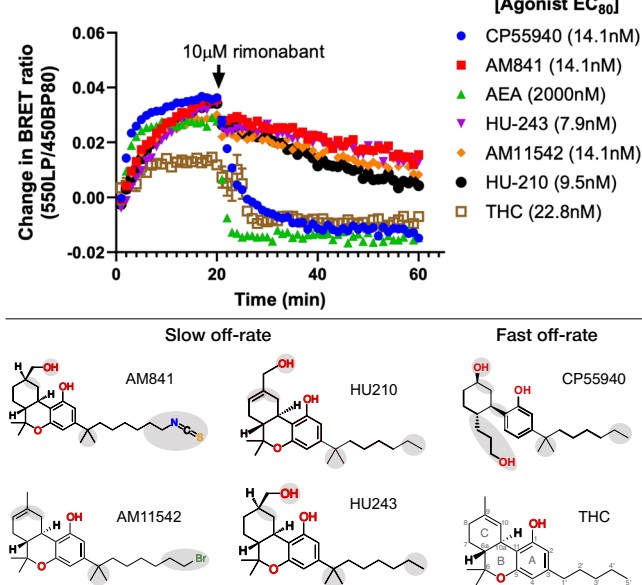

**Fig. 6 | Structure-kinetics relationships of THC analogs studied for reversibility of agonist activity (mG$_i$ recruitment) upon addition of the CB$_1$ antagonist rimonabant.** The plot shows the change in basal BRET upon addition of agonist (0 min, EC$_{80}$ concentration) and rimonabant (20 min, 10 μM). Data from HEK293-TR cells stably expressing CB$_1$-Nluc and NES-venus-mG$_i$. The number of observations (*n*) is 6 for each of the individual ligand dissociation curves except for THC (*n* = 5). Data plotted is mean ± S.E.M. THC as a partial agonist has a relatively small response; therefore, the visible error bars correspond to an "addition artifact" which is more apparent for this ligand.

HU210[43,44] (Supplementary Fig. 6). We found that G$_i$ recruitment and cAMP inhibition correlate very well with both E$_{max}$ and pEC$_{50}$ values (r² values of 0.99 and 0.85, respectively). Thus, this suggests that G protein recruitment does indeed correlate with the strength of cellular signaling outputs in the CB$_1$ receptor system when activated by physiological ligands as well as THC analogs.

**Low-potency/low-efficacy THC analogs exhibit recruitment bias for G proteins over arrestin**

To assess if THC analogs display different activity also in terms of biased signaling, we complemented the G$_{i1}$ recruitment with arrestin recruitment experiments. By generating bias plots (Supplementary Fig. 7), we find that all ligands show a stronger recruitment of mG$_i$ than arrestin. This indicates a general difference in the sensitivity of the two systems, whereas ligand bias can only be defined relative to a reference ligand[45]. We therefore calculated recruitment bias of tested ligands relative to two reference ligands, the endogenous ligand 2AG and THC (Supplementary Data 1, MS Excel file). Interestingly, 2AG has low potency/high efficacy and THC has high potency/low efficacy (Fig. 3), but these ligands are nearly unbiased relative to each other because

their opposite efficacy/potency relationships equal out when calculating log(E$_{max}$/EC$_{50}$) values. Specifically, their relative bias factor (fold-difference in log(E$_{max}$/EC$_{50}$) across ligands and pathways) is a moderate 1.3 in both cases, for arrestin (THC) or G$_{i1}$ (2AG).

The group of low-potency/low-efficacy THC analogs (Fig. 2c) shows either very weak or no arrestin recruitment. For example, in the case of JWH133 its arrestin E$_{max}$ was estimated to be 9.1% of the 2AG response, while the response to L759633 was too low to fit a curve and Cannabinol did not show any arrestin recruitment. Our observation that only low-potency/low-efficacy THC analogs display recruitment bias indicates that more chemical diversity would be required to achieve strong bias while maintaining potency and efficacy.

**Structure-kinetics relationship**

We conducted a series of kinetics experiments designed to benchmark the apparent $k_{off}$ values of THC and five analogs differing structurally in their tricyclic ring, alkyl branch, and alkyl tail (Fig. 6, Supplementary Fig. 8 and Supplementary Table 4). These experiments show the relative differences in ligand dissociation on mG$_i$ binding following application of a saturating concentration of the CB1R-specific inverse agonist rimonabant. We find that AM841, AM11542, HU210, and HU243 exhibit slow off-rates from the activated CB$_1$ whereas THC, CP55940, and the endogenous CB1R agonist AEA instead show relatively fast off-rates. This is in agreement with previous studies demonstrating that the slow off-rate of these ligands is independent of an isothiocyanate moiety and covalent bond[16,46] and that some ligands in this series have fast off-rates[47]. All four ligands with slow dissociation share the tricyclic ring, alkyl branch, and alkyl tail with seven carbons. Given that CP55940 is the only ligand lacking the B-ring its difference in kinetics is likely caused by an increased conformational freedom or reduced bulk at the core scaffold. In MD simulations, CP55940 has an increased flexibility in this region destabilizing key hydrophobic interactions. For THC, the fast off-rate can instead be attributed to the unique shorter,

five-carbon alkyl tail. As THC has a lower efficacy than its studied analogs it displays a smaller total drop in BRET ratio, whereas the steepest drop in binding is observed for the endocannabinoid AEA. Although its long aliphatic chain is similar to the C3 alkyl tail of THC analogs, AEA lacks the tricyclic scaffold fitting tightly into the binding pocket and making many hydrophobic interactions. Altogether, the kinetics experiments demonstrated that the tricyclic moiety, alkyl branch (1′,1′-dimethyl), and seven-carbon alkyl tail are essential features for THC analogs to bind to $CB_1$ in a slow off-rate, "wash-resistant" manner.

## Discussion

Structural biology on GPCRs is approaching a throughput that allows comparison of complexes of close ligand analogs, especially when, like here they are complemented with manual docking or ligand substitution. Furthermore, high-performance computing has transformed molecular dynamics simulations, enabling the exploration of complex interactions involving multiple ligands bound to fully activated receptors, which in turn are bound to their cognate G proteins e.g.,[48,49]. A combination of these techniques allowed us to delineate detailed structure-activity relationships at the level of spatiotemporal ligand-receptor interactions. Moving beyond a solely static ligand-based analysis to one that encompasses receptor-effector complex explorations unlocks the potential for identifying and exploiting previously unexplored proximal binding sites, thus broadening the scope of therapeutic intervention and drug design strategies. Furthermore, it provides a deeper, target-complementarity rationale for favorable pharmacophore elements informing the optimal bioisosteric replacements. Exciting prospects are currently opening for such rational, structure-based drug design from recent breakthroughs demonstrating highly accurate modeling of ligand-protein complexes using AlphaFold[50] or RoseTTAFold All-Atom[51]. Finally, to attain the structural basis for design of better drugs through biased signaling, it would be highly desirable to determine complexes of differentially biased ligands bound to the same receptor and their respective preferred cognate transducer protein.

Our structure-activity relationship study covers diverse structural analogs of THC with modifications in key pharmacophore features and consistent pharmacological evaluation of potency, efficacy, and signaling bias for $G_{i/o}$ and β-arrestin. Our analysis adds to previous studies by plotting ligands by potency and efficacy (Fig. 3), allowing separation of ligands and determinants by both pharmacological properties and a comprehensive analysis of ligand-receptor binding modes. This took advantage of the new high-resolution/-coverage $CB_1/G_{i1}$ structure providing a consistent receptor, while we could also preserve ligand conformations from recent experimental structures, where available[16–18]. The molecular dynamics simulations could explain differential activities not readily understood from ligand structures or static binding modes alone. For example, THCv only differs from THC by a two-carbon alkyl tail reduction but displays dramatic inverse agonism and low potency. Here, MD revealed a highly "wobbly" binding causing the receptor interactions to become unstable and infrequent. Another point in case of MD is that although preserving a C1 hydroxyl, Cannabinol—like the non-hydroxyl analogs JWH133 and L759633—loses the hydrogen bond to S383$^{7x38}$. This observation provides a structural basis contributing to Cannabinol's relatively low affinity and potency at $CB_1$ and to JWH133's and L759633's $CB_2$ selectivity.

These studied compounds vary in alkyl tail length (shorter in THC) and modifications (isothiocyanate in AM841 and bromine in AM11542), B-ring (open in CP55940), and C1′ alkyl branch (missing in THC). Furthermore, we included the endocannabinoid AEA as an additional, non-THC-analog reference ligand. Our results demonstrate that the length of the alkyl tail and completeness of the tricyclic ring moiety are structural determinants for dissociation kinetics from the active form

of the receptor. This corroborates a recent study that performed washout experiments of 14 compounds similar to AM841[46]—all of which had a heptyl or octyl tail and displayed wash-resistant binding. The tight binding which results in slow dissociation, is likely due to the ligands' fit in the binding pocket and molecular interactions, the majority of which are hydrophobic. The early belief that AM841 binds irreversibly due to a covalent bond was based on the observation that mutations of C355$^{6x47}$ to serine, alanine, and leucine significantly increased radioligand binding relative to wildtype $CB_1$ in the presence of AM841[52]. This has now been disproven by AM841-$CB_1$ structure complexes[16,17] and recent washout studies[16,46]. Of note, C355$^{6x47}$ is part of a highly conserved C355$^{6x47}$WxP motif in TM6 involved in ligand binding and activation of class A GPCRs[53]. Hence, the most plausible explanation of the mutagenesis effects is through weakening of ligand binding and receptor activation.

The THC analogs HU210 and CP55940 have been previously reported to be biased towards the $G_{i/o}$ family (cAMP inhibition) over phosphorylation of extracellular signal-regulated kinase 1/2 (pERK1/2)[43]. However, pERK1/2 is a downstream protein with activity that is dependent on both G proteins and arrestins. Therefore, its bias cannot be delineated directly at the level of the receptor-binding transducers. Our experiments separate these pathways and show that only low-potency/low-efficacy THC analogs have a strong bias towards G protein over arrestin recruitment, which is weak or abrogated (Supplementary Fig. 7 and Supplementary Table 2). This indicates that more chemical diversity would be required to achieve strong bias while maintaining potency and efficacy. Several compound classes activating $CB_1$ have already been found to exhibit biased signaling, including endocannabinoids, phytocannabinoids, synthetic cannabinoids, indoles, and biphenylureas[13]. Furthermore, in our experiments, THC displayed a slight preference (bias factor 1.3) for arrestin over $G_{i1}$ compared to the endogenous agonist 2AG. All biased signaling experiments are heavily system-dependent and these experiments, performed on transducers directly binding to the receptor, often do not translate to in vivo settings[45]. For example, both THC and 2AG have been shown to be arrestin-selective over $G_{i/o}$ in a cell model of striatal neurons[54]. Together, this warrants many more biased signaling studies covering different $CB_1$ ligand scaffolds across the transducer, downstream pathway, and in vivo levels. For example, to investigate if biased ligands can be identified that induce therapeutic effects, such as analgesia, without common side effects e.g., hypothermia, catalepsy, and hypolocomotion[55].

Taken together, the HU210/$CB_1$/$G_{i1}$ cryo-EM structure, pharmacological profiling, ligand-receptor docking, and molecular dynamics simulations link functional groups of THC ligands to spatiotemporal interactions determining efficacy, potency, biased signaling, and kinetics. These structure-activity relationships serve to guide further development of THC analogs as tool compounds and therapeutics.

## Methods
### Protein expression
Human cannabinoid receptor 1 (CB1) residue 2-472 was fused with a N-terminal hemagglutinin (HA) signal peptide followed by a Flag tag and a short Gly-Ser linker. At its C-terminus, the receptor was fused to a 10-histidine purification tag. The DNA sequence was codon optimized for expression in insect cells and inserted into the pFastBac1 vector using custom DNA synthesis (Genscript) for Bac-to-Bac virus expression. A dominant negative mutant of human GNAi1 (G203A/A326S)[37,56] was also inserted into PfastBac1. Viruses co-expressing human GNB1 and GNG2 together, as well as Ric8A were kindly provided as a gift by Daniel Hilger. The $CB_1$ receptor $G_i$ protein complex was formed by co-expression in *Spodoptera frugiperda* (*Sf*9) insect cells. Cells were infected at 2 mio/ml cells with virus at a ratio between receptor, GNAi1, GNB1/GNG2, and Ric8A at (8:5:2:1). Cells were harvested by centrifugation after 48 h and frozen before subsequent purification.

A Fab fragment (scFv16) construct essentially identical to the construct reported in ref. 57 was expressed using pFastBac1 in BTI-Tn-5B1-4 (High Five/Hi5) cells. Cells were infected at 2 mio/ml cells and harvested after 48 h. Cells were removed by centrifugation and the supernatant containing the secreted scFv16 protein was frozen with 10% glycerol.

## ScFv16 purification

ScFv16 was purified using HisPur™ Ni-NTA resin (Thermofisher sci. Cat. No. 88221). In short, initially, impurities were removed by precipitation by adding 30 mM Tris pH 8.0, 1 mM $NiSO_4$, and 5 mM $CaCl_2$ for 1 h at room temperature followed by centrifugation at 18,000 rpm for 20 min. The supernatant was filtered through a 0.22 μM filter and loaded onto Ni-NTA resin overnight at room temperature. The resin was then washed in wash buffers: Wash buffer 1:20 mM Hepes pH 7.4, 500 mM NaCl and 10 mM imidazole, wash buffer 2:20 mM Hepes pH 7.4, 100 mM NaCl and 10 mM imidazole, and wash buffer 3: 20 mM Hepes pH 7.4, 100 mM NaCl and 30 mM imidazole. The protein was eluted with a buffer containing 20 mM Hepes pH 7.4, 100 mM NaCl, and 300 mM imidazole. The histidine tag of the protein was removed by digesting with 3 °C protease for 3 days at 4 °C. ScFv16 was concentrated to -1.5 mg/ml and frozen with 20% glycerol.

## Complex formation and purification

The cell pellet from 1 L of media containing the $CB_1$ $G_i$ protein complex was lysed at room temperature for 1 h with stirring. The lysis buffer consisted of: 20 mM Hepes pH 7.4, 50 mM NaCl, 4 mM $MgCl_2$, 2 μM HU-210 (Tocris Cat. No. 0966), -10 μg/ml ScFv16, 0.025 units/ml Apyrase (NEB Cat. No. M0398S) and Roche EDTA free protease inhibitors (Sigma Cat. No. REF05056489001). The lysed cells were harvested by centrifugation for 20 min at 11,000 rpm. The complex was then solubilized using a 40 ml dounce tissue grinder with 30 strokes and a solubilization buffer containing 20 mM Hepes pH 7.4, 100 mM NaCl, 4 mM $MgCl_2$, 1 μM HU-210, -20 μg/ml ScFv16, 0.05 units/ml Apyrase (NEB Cat. No. M0398S), 15% glycerol, 0.5% LMNG (Anatrace Cat. No. NG310), 0.03% Cholesteryl hemisuccinate (Sigma Cat. No. C6512-25G) and Roche EDTA free protease inhibitors. The sample was mixed for 2 h at 4 °C. Non solubilized material was removed by centrifugation for 40 min at 14.000 rpm. 5 mM $CaCl_2$ and 0.75 ml M1 anti flag resin added to the supernatant. The slurry was gently mixed for 1 h at 4 °C and the beads with bound complex were collected by centrifugation for 5 min at 1000 g and transferred to a 1.5 ml filter column (Bio-Rad Cat. No. #7311550EDU). The resin was washed with purification buffer containing 20 mM Hepes pH 7.4, 100 mM NaCl, 5 mM $CaCl_2$, 2 mM $MgCl_2$, 500 nM HU-210, 0.01% LMNG and 0.006% Cholesteryl hemisuccinate. The protein was eluted in purification buffer supplemented with 10 mM EDTA and 200 μg/ml Flag peptide (Sigma Cat. F3290-4MG). The eluted protein was supplemented with an additional 1 μM HU-210 and concentrated to -50 μl using 100 kDa spin concentrators (Merck cat. nr. Z614092) at 1000 g. Subsequently, the complex was frozen in liquid nitrogen. Immediately before preparation of grids for cryo-EM, the sample was run on a Superose 6 Increase 3.2/300 column on a Äkta system using purification buffer for elution at 0.05 ml/min.

## Cryo-EM

Prior to sample application the cryo-EM Quantifoil R 1/1 C 300 grids were activated using a Gloqube Plus glow discharger at 15 mA for 45 s. All grids were prepared using a Leica EM GP2 automatic plunge freezer. three microliters of complex at -0.85 mg/ml were loaded using a total blotting time of 6–8 s.

## Cryo-EM data collection

Data collection was performed on a Titan Krios G3i (Thermo Fisher Scientific) with a K3/BioQuantum detector/energy filter setup (Gatan).

Magnification was set at 130kx, resulting in a physical pixel size of 0.647 Å/px. Automated data collection was done using EPU (Thermo Fisher Scientific) set to collect in super-resolution with 2x binning to the physical pixel size, with gain correction on the fly. Aberration-free image shift (AFIS) was used for faster data collection speed with 1 exposure per hole for 11260 movies total. Defocus targets were set from −0.6 to −1.8 μm in steps of 0.2 μm, with autofocus after distance at 6 μm. A 50 μm C2 aperture and no objective aperture were used. The energy filter was tuned and set to a slit width of 20 eV with auto-centering of the zero-loss peak (ZLP) every hour. An exposure time of 1.50 s in 56 frames for 60.6 e/$Å^2$ total dose was employed.

## Cryo-EM data processing

CryoSPARC Live (Structura Biotechnology[58]) was used to monitor data quality during the data collection and provided an initial 3D volume (Supplementary Fig. 9). The data was pre-processed in Relion 3.1[59] followed by particle picking using crYOLO[60] resulting in 603k particles (Supplementary Fig. 10). The particle stack was cleaned by iterating between jobs of 3D Classification and 3D Auto-refine using Sidesplitter[61], ending with a polished stack of 187k particles (Supplementary Figs. 11–12). The stack was imported to CryoSPARC and subjected to non-uniform 3D refinement, followed by a Local Refine job with a mask around the TM region resulting in a 2.9 Å overall resolution (Supplementary Figs. 13–14).

## HU210/$CB_1$/$G_{i1}$ structure model building

The published structure of $CB_1$ bound to the ligand AM841 PDB: 6KPG [17] was used as the starting model for our refinement. The starting model for the alpha-helical domain of the $G_{i1}$ subunit was retrieved from the crystal structure of the scFv16 bound $G_{i1}$ hetero-trimer PDB: 6CRK [57]. Model building and refinement were done using the Coot[62] and Phenix[63] software packages (Supplementary Figs. 15 and 16).

**THC analog – $CB_1$ structure/model preparation.** The HU210/$CB_1$/$G_{i1}$ structure was prepared for THC analog docking and molecular dynamics simulations with THC and analogs. We removed ScFv16, reverted the two $G\alpha_{i1}$ mutations (G203A and A326S) to wildtype, and modeled in the missing ICL3 (residues 316-334) of $CB_1$ receptor using MODELER[64]. The structure was prepared using the Protein Preparation Wizard tool[65] and OPLS3e force field[66] in Schrödinger[67]. For, AM841[17], AM11542[16], and CP55940[18] conformations were taken from their respective $CB_1$ structure complexes (the latter two lacking a G protein) and superposed to HU210 to generate $CB_1$/$G_{i1}$ complexes. For the remaining THC analogs, $CB_1$/$G_{i1}$ complexes were constructed by using the Maestro 3D Builder Tool[67] to edit HU210 in our $CB_1$/$G_{i1}$ structure and perform energy minimization.

## Molecular dynamics simulations

Protonation states of ionizable residues were assigned using Epik[67,68] using a pH value of 7.0 as reference. Restrained minimization was performed using the OPLS3e force field[66]. A membrane was built and positioned around the receptor with guidance from the PPM 2.0 Web Server of the Orientations of Proteins in Membranes (OPM) database[69]. An orthorhombic box shape with 11 Å buffer distance in the x, y, and z directions was used to build a solvent box around the membrane-bound agonist-receptor-transducer complexes. All-atom 1-palmitoyl-2-oleoylphosphatidylcholine bilayer (POPC) and TIP3P water were used for the lipid and water models, respectively. A salt concentration of 0.15 M was set, and all the systems were neutralized using sodium and chloride ions.

All MD simulations were performed using the Desmond Molecular Dynamics System[70] with the OPLS3e force field[66] and full particle mesh Ewald electrostatics[71]. The systems were gradually heated to 300 K in

the NVT ensemble and allowed to equilibrate for 50 ns. Production MD simulations at constant temperature (300 K) and pressure (1 atm) were performed in triplicates of 1000 ns simulation time for each system. This resulted in a total sampling of 3 µs for each of the 11 receptor-ligand systems. Analysis of MD trajectories to calculate frequencies of receptor-ligand and intra-receptor contacts was done using the Get-Contacts MD analysis package[72]. Root mean square deviation (RMSD) calculation was performed using the RMS calculation tool of GROMACS 2024.3[73,74] (Supplementary Fig. 17). Interatomic distances were calculated using the MD Analysis package[75,76] (Supplementary Fig. 18). Chemical structures were drawn with Marvin version 24.3.1, ChemAxon (http://www.chemaxon.com) and ChemDraw version 23.1.1. Generic residue numbers (GRNs) of the $CB_1$ receptor residues were referenced using the "Generic residue number tables" resource in GPCRdb[77,78].

**Cell culture and molecular biology reagents.** The HEKT-REx™-293 cell line (Cat. No. R71007) was purchased from Invitrogen (CA, USA). T75 mammalian cell culture flasks were purchased from Fisher Scientific (Loughborough, UK). Cell culture reagents from Sigma Aldrich (St. Louis, MO, USA) include Dulbecco's Modified Eagle's Medium (DMEM) −high glucose (Cat. No. D6429), Dulbecco's Phosphate Buffered Saline (D-PBS, Cat. No. D8537), Hank's Buffered Saline Solution (HBSS, Cat. No. H8264), 4- (2-hydroxyethyl)-1-piperazineethanesulfonic acid sodium salt (HEPES, Cat. No. RDD035-100G), bovine serum albumin (FBS, Cat. No. F7524), trypsin/EDTA solution 100 mL (Cat. No. R001100). Reagents purchased from Gibco™ (MA, USA) included Blasticidin™ Selection Reagent HCl 10 mg/mL (Cat. No. 12172530), Zeocin™ Selection Reagent 100 mg/mL (cat. No. R25005), Geneticin™ Selective Antibiotic (G-418 Sulfate) 50 mg/mL (Cat. No. 10131035). Reagents from Corning® (Corning, NY, USA) include Corning® 100 mL Cellstripper™, liquid (Cat. No. 25-056-Cl). Polyethylenimine (PEI, Cat. No. 23966-1) was obtained from Polysciences Inc (PA, USA). 96-well cell culture plates (Cat. No. 655098) were purchased from Greiner Bio-One (Stonehouse, UK). The expression vectors pcDNA™4/TO were obtained from ThermoFisher Scientific and pcDNA™ 3.1 from Invitrogen™.

**Compounds.** Reference ligands were obtained from Bio-Techne® Tocris (Abingdon, Oxfordshire, UK) include HU210 or ((6a*R*)-*trans*-3-(1,1-Dimethylheptyl)-6a,7,10,10a-tetrahydro-1-hydroxy-6,6-dimethyl-6*H*-dibenzo[*b,d*]pyran-9-methanol) (Cat. No. 0966), anandamide (AEA) or *N*-(2-Hydroxyethyl)-5*Z*,8*Z*,11*Z*,14*Z*-eicosatetraenamide (Cat. No. 1339), 2-Arachidoylglycerol (2AG) or (5*Z*,8*Z*,11*Z*,14*Z*)-5,8,11,14-Eicosatetraenoic acid, 2-hydroxy-1-(hydroxymethyl)ethyl ester (Cat. No. 1298), and Rimonabant (SR-141716A) or (*N*-(Piperidin-1-yl)-5-(4-chlorophenyl)-1-(2,4-dichlorophenyl)-4-methyl-1*H*-pyrazole-3-carboxamide hydrochloride (Cat. No. 0923), (-) Cannabinol or 2-[(1 R,6 R)-3-Methyl-6-(1-methylethenyl)-2-cyclohexen-1-yl]-5-pentyl-1,3-benzenediol (Cat. No. 1570), JWH133 or (6aR,10aR)-3-(1,1-Dimethylbutyl)-6a,7,10,10a-tetrahydro-6,6,9-trimethyl-6H-dibenzo[b,d]pyran (Cat. No. 1343/10). CP55940 or (5-(1,1-Dimethylheptyl)-2-[5-hydroxy-2-(3-hydroxypropyl)cyclohexyl]phenol) (Cat. No. C1112), Nabilone (Cat. No. N3785), Tetrahydrocannabivarin (THCv) (Cat. No. T-094) and Δ⁹-Tetrahydrocannabinol solution (THC, Cat. No. T2386) was obtained from Merck. L759633 (Cat. No. CAY10009280-1 mg) was obtained from Cambridge Biosciences.

**Cell culture**

Cultured cells were maintained in a humidified incubator at 37 °C and 5% $CO_2$ in Dulbecco's modified Eagle's medium (DMEM) (Sigma-Aldrich) containing 10% fetal bovine serum (FBS). Stable cell lines expressing both the $CB_1$ plus venus-mGsi, and the $CB_1$ plus venus-β-arrestin2, were created in T75 flasks, using a 3:1 ratio of polyethylenimine (PEI): DNA. 24 h post transfection cells were split into

T175 flasks and maintained as stable cell lines. Antibiotics including blasticidin (5 µg/mL) and Zeocin™ (20 µg/mL) were introduced as selection agents to create stable cell lines expressing the pcDNA™4/TO a mammalian expression vector that encodes the appropriate $CB_1$ sequence (ThermoFisher Scientific, UK). Whilst Geneticin® (G-418 sulphate, 200 µg/mL) was used to select for cells containing the pcDNA™ 3.1 mammalian expression vector (Invitrogen™), encoding venus-mGsi and venus-β-arrestin2. A mixed population of stable cells was eventually produced with cells having resistance to the selection agents employed.

**Mini-G protein and β-arrestin recruitment assays in $CB_1$-expressing cells.** $CB_1$ coupling to G proteins was assessed using a fluorescent G protein surrogate, venus-mini-Gsi1 (vmGsi) protein[79]. The venus-mGsi subunit is essentially a chimeric protein consisting of C-terminal $G_{i1}$ residues grafted onto venus-mGs and originally engineered from the native Gsa protein. The venus-mGsi is ideal for studying receptor activation of the $CB_1$, as unlike its wild-type Gai1 counterpart, the resulting active-receptor complex formed is stable and resistant to nucleotide exchange meaning that active state signaling is maintained if the agonist is present.

HEK293TR-CB₁-nLuc cells expressing fluorescently labeled miniG or β-arrestin protein were maintained in a humidified environment at 37 °C and 5% $CO_2$ in Dulbecco's modified Eagle's medium (DMEM) with 10% fetal bovine serum (FBS) containing blasticidin (5 µg/ml), Zeocin (20 µg/ml) and G418 (0.2 mg/mL) and used to assess compound stimulated β-arrestin recruitment to the human $CB_1$. Cultured cells were harvested upon reaching 70% confluency and plated at a seeding density of 50,000 cells per well in poly-D-lysine (5 µg/mL) coated clear-bottomed 96-well cell culture plates. The cells were grown for 48 h until they reached confluency and then stimulated with (1 µg/mL tetracycline) for a further 48 h. Media was aspirated, and the cells were washed in 100 µL/well PBS, then 90 µL/well assay buffer (HBSS, 0.5% BSA, 5 mM HEPES) with 10 µM furimazine was applied to each well. A white back seal was applied to plates, which were then incubated for 15 min at 37 °C to allow the furimazine to enter the cells. Assay plates were then transferred to the PHERAstar FSX set to a temperature of 37 °C, and three BRET cycles were run to collect an initial baseline reading, after which 10 µL of compounds diluted in assay buffer was added to the plate, and the plate was read at 1-min intervals for 30 min. Compounds were serially diluted in DMSO, before a 1/10 dilution in assay buffer and a further 1/10 dilution on addition to the assay plate. Buffer containing 10% DMSO (1% final) served as the vehicle control with all responses normalized to the maximal response produced by 2AG. Mini-g protein reversal experiments were performed essentially as described above, by applying a concentration of agonist producing 80% of its own maximum response ($EC_{80}$) to cells in a 96-well cell culture plate, followed by the addition of an excess of rimonabant (10 µM) or vehicle, with the resulting BRET signals monitored for up to 90 min.

**Signal detection and data analysis**

Raw experimental mGi and β-arrestin2 recruitment data were collected at 1 min intervals on the BMG PHERAstar FSX (BMG Labtech, Offenburg, Germany), and processed using MARS data analysis software (BMG Labtech), as the ratio of BRET 1 (535-30LP/475-30BP). This data was then exported in Microsoft Excel and transferred to GraphPad PRISM 9.2 (GraphPad Software, San Diego, U.S.A.). A kinetic analysis of compound-induced response measured over time was completed by plotting the resulting reported BRET ratios. Characterization of agonist CBR responses was achieved by selecting the concentration-response data at a fixed time point and one producing the maximal observable responses to the ligands under test. Concentration-response data was then normalized to the reference ligand 2AG. Individual concentration-response data were fitted to sigmoidal (variable

slope) curves using a "four-parameter logistic equation":

$$Y = Bottom + (Top - Bottom)/(1 + 10^{(LogEC_{50} - X)*Hillslope}) \quad (1)$$

Where Bottom and Top are the plateaus of the agonist and inverse agonist concentration-response curves. $LogEC_{50}$ is the concentration of agonist/inverse agonist that gives a half-maximal effect, and the Hillslope is the unitless slope factor. Individual agonist $EC_{50}$ and $E_{max}$ values are reported as the Mean ± SEM, from the number (n) of individual experiment indicated.

All normalized data of the individual concentration points from each individual experiments were pooled, and bias plots were constructed by means of a "centered second-order polynomial" fitting of the normalized and pooled data obtained for the individual concentration points of the mini-$G_i$ (x-coordinate) and β-arrestin2 (y-coordinate) assay formats.

The rates of mini-$G_i$ protein reversal were estimated for each agonist in Prism 9.0 using the following equation which describes a 'one phase exponential decay':

$$Y = Span*exp^{(-k_{off}*X)} + Plateau \quad (2)$$

Individual agonist $k_{off}$ values are reported in tables as the Mean ± SEM, from the number (n) of individual experiment indicated.

**Compound synthesis.** *(6aR,9 R,10aR)-9-(hydroxymethyl)-3-(8-isothiocyanato-2-methyloctan-2-yl)-6,6-dimethyl-6a,7,8,9,10,10a-hexahydro-6H-benzo[c]chromen-1-ol (AM841).* To a stirred solution of (6aR,9 R,10aR)-3-(8-azido-2-methyloctan-2-yl)-9-(hydroxymethyl)-6,6-dimethyl-6a,7,8,9,10,10a-hexahydro-6H-benzo[c]chromen-1-ol (56 mg, 0.13 mmol) in anhydrous tetrahydrofuran (2.6 mL) at room temperature under an argon atmosphere were added triphenylphosphine (170 mg, 0.65 mmol) followed by carbon disulfide (230 μL, 3.9 mmol) and the mixture was stirred at that temperature for 24 h. Upon completion, the volatiles were evaporated under reduced pressure and the residue was purified by flash column chromatography (silica gel; 25% ethyl acetate in hexanes as eluent) to afford 49 mg of AM841. Physical, analytical, and spectroscopic data were identical to those we reported earlier[16].

*(6aR,10aR)-3-(8-bromo-2-methyloctan-2-yl)-6,6,9-trimethyl-6a,7,10,10a-tetrahydro-6H-benzo[c]chromen-1-ol (AM11542).* To a mixture of 5-(8-bromo-2-methyloctan-2-yl)benzene-1,3-diol (124 mg, 0.39 mmol) and p-toluenesulfonic acid (14 mg, 0.08 mmol) in anhydrous methylene chloride (3 mL) at 0 °C under an argon atmosphere was added a solution of (4 R)-1-methyl-4-(prop-1-en-2-yl)cyclohex-2-en-1-ol (71 mg, 0.47 mmol) in anhydrous methylene chloride (1 mL) and the resulting mixture was stirred at that temperature for 40 min. Upon completion, the reaction was quenched by saturated aqueous sodium bicarbonate, and the organic layer was dried over sodium sulphate and evaporated. The residue was purified by flash column chromatography (silica gel; 5% diethyl ether in hexanes) to afford 127 mg of (1′R,2′R)-4-(8-bromo-2-methyloctan-2-yl)-5′-methyl-2′-(prop-1-en-2-yl)-1′,2′,3′,4′-tetrahydro-[1,1′-biphenyl]-2,6-diol. The product (122 mg, 0.27 mmol) was dissolved in anhydrous methylene chloride (8 mL) under an argon atmosphere and cooled to 0 °C. Boron trifluoride etherate (160 μL, 1.3 mmol) was added and the mixture was stirred for 30 min at 0 °C followed by 6 h at ambient temperature. The reaction was quenched by saturated aqueous sodium bicarbonate and the organic layer was dried over sodium sulphate and evaporated. The residue was purified by flash column chromatography (silica gel; 5% ethyl acetate in hexanes) to afford 105 mg of AM11542. Physical, analytical, and spectroscopic data were identical to those we reported earlier[80].

**Reporting summary**
Further information on research design is available in the Nature Portfolio Reporting Summary linked to this article.

## Data availability
The HU210/$CB_1$/$G_{i1}$ cryo-EM structure and density are available in the Protein Data Bank under accession code 9ERX. The corresponding cryo-EM map has been deposited in the Electron Microscopy Data Bank (EMDB) under accession code EMD-19929. The source data underlying Figs. 2a–e and 6 (MS Excel), and initial and final configurations of one MD trajectory per ligand (PDB files), are provided as a Source Data file. Source data are provided with this paper.

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

## Acknowledgements

We thank Milena Timcenko for advice on sample preparation towards cryo-EM. We thank Dr. Shan Jiang for contribution to the exploration of synthetic methodologies. This work was funded by grants from the Lundbeck Foundation (R313-2019-526 and R383-2022-306) and Novo Nordisk Foundation (NNF18OC0031226) to D.E.G. D.E.G., J.S.K., M.G. and T.S.T. are members of the Integrative Structural Biology at the University of Copenhagen (ISBUC) cluster. P.H. was funded by an international fellowship awarded by the Office of Educational Affairs, Thailand. D.A.S and D.B.V gratefully funding by acknowledge Roche Postdoctoral Fellowship RPF-551 funding by F. Hoffmann-La Roche Ltd., Basel, Switzerland and Medical Research Council (MR/Y003667/1). A.M. acknowledges funding from NIDA (DA009158). P.N. acknowledges support from a Distinguished Investigator grant from the Novo Nordisk Foundation (NNF19OC0054875) and a Lundbeck Foundation professorship grant (R310-2018-3713). Support is acknowledged for the Danish national cryo-EM facility EMBION with infrastructure grants from the Danish Ministry for Research and Higher Education (5072-00025B), and the Novo Nordisk Foundation (NNF20OC0060483).

## Author contributions

T.S.T. produced the HU210/CB$_1$/G$_{i1}$ samples and grids supervised by J.S.K. and M.G. A.B. and T.D. made the cryo-EM data collection, processing, and map reconstruction under supervision of T.B. and P.N. T.S.T. and J.S.K. performed the model building and refinement of the structure. D.A.S. and P.H. performed all pharmacological experiments supervised by D.B.V. Y.K. performed all computational and structure-activity relationship analyses and drafted the manuscript together with D.E.G. C.I.T. and S.P.N. synthesized AM841 and AM11542 under the supervision of and A.M. H.D. made a literature review and initial analysis of ligand binding kinetics (all work done before employment at H. Lundbeck A/S). Y.K. and D.A.S. produced the illustrations. D.E.G. conceptualized and managed the project. All authors have approved the submitted manuscript.

## Competing interests

D.E.G. is a part-time employee and warrant-holder at Kvantify. D.A.S and D.B.V. are both founders and directors of Z7 Biotech Ltd, an early-stage drug discovery CRO. The remaining authors declare that the research was conducted in the absence of any commercial or financial relationships that could be construed as a potential conflict of interest.
