## [Transparent Peer Review file · Nature Communications]

Structural basis of Δ 9-THC analog activity at the Cannabinoid 1 receptor

Corresponding Author: Professor David Gloriam

Version 1:

Reviewer comments:

Reviewer #2

(Remarks to the Author)

The manuscript titled "Structural basis of delta 9 -THC analog activity at the Cannabinoid 1 receptor" describes the cryo-EM structure of HU210 bound to CB1R and its primary transducer Gi1. A series of docking and molecular dynamics simulations on THC and its analogs were conducted and pharmacological profiles were constructed to compare affinities, kinetics, and agonist biases.

This is not the first reported structure of a CB1R-ligand complex, but the HU210/CB1/Gi1 complex represents the largest coverage of receptor residues reported to date and moreover, this is the first disclosure of THC analog dissociation rates from an active CB1 receptor complex under experimental conditions. The methodology is sound, and there is sufficient detail provided in the methods for the work to be reproduced.

The graphical representation and data analysis is noteworthy (particularly Figure 3 and 4). The pharmacological grouping based on efficacy/potency derived from experimental data is clear.

Minor Concerns:

The structure-kinetics relationship section describing AM841 to CB1R (p.6, lines 246-259) is confusing. The authors discuss the contradictory nature of the interaction between AM841 and CB1R based on several published reports (covalent vs. non covalent) and continue this discussion (p.8, lines 319-325). If this has been settled (line 322, "This has now been disproven by AM841-CB1 structure complexes"), then why spend time in a discussion? If this has not been settled, why haven't the authors considered reversible covalent inhibition (which is observed routinely with isothiocyanates). If this compound is a point of contention, why include it as one of the 10 analogs?

I am not convinced that the statement "This may advance our understanding of fundamental receptor function and support the further development of THC and its analogs as therapeutics." is necessary.

It is important to specify delta9 -THC and analogs vs. delta8 -THC and analogs. Although delta8 -THC is indeed an analog of delta9 -THC, specific designations are helpful to avoid ambiguity. For example, Fig. 2 and Fig. 3 (delta-9 THC analogs) and the Extended Data Fig. 1 (p. 22) is titled "Chemical structures of delta9-THC analogs and additional agonists assayed at CB1". There are a number of delta8 -THC analogs illustrated in these figures (including the main compound studied, HU210). Furthermore, it may be beneficial to edit the title to reflect the concerns listed above.

Reviewer #3

(Remarks to the Author)

I have reviewed the manuscript by Thor S. Thorsen et al., which investigates the cryoEM structure of the Gi1-bound CB1 receptor with HU210. The study provides a comprehensive pharmacological profiling of this receptor, comparing its efficacy

and potency with 12 known analogues, including 10 exogenous and 2 endogenous ligands. The authors employ BRET techniques to compare biased signaling and kinetics and utilize all-atom molecular dynamics simulations over 1000 ns to elucidate the structure-activity relationship of the exogenous analogues. Then atomic level pharmacological explanations described based on the contact frequency between binding pocket residues and ligands.

This work represents a valuable contribution to the understanding of GPCR structure and pharmacology. However, several concerns need to be addressed before the manuscript can be considered for publication. You can find comments below:

1. Although the authors conducted 1000 ns all-atom molecular dynamics simulations, they did not present detailed data or results from these simulations. First, the methods section should specify the number of replicates for each system (each drug) of MD simulation.
2. The reproducibility of the system should be demonstrated, ideally in the supplementary materials, using RMSD data for CB1/Gi1 with drug complexes throughout the simulation time. Given the complexity of GPCR membrane systems and the relatively less-sampled conformational dynamics, this potentially be affect to results of interactions analysis between comparisons of structurally similar analogous compounds and binding pocket residues
3. The description of the spatiotemporal interactions between the THC analog and the CB1 receptor, as shown in Fig. 4c, is insufficient. The authors should include data on contact frequency throughout the simulation time to provide a more comprehensive understanding and confidence.
4. It would be beneficial if the authors could provide time series data for hydrogen bond analysis, particularly for Fig. 5(b), 5(f), and 5(g).
5. The manuscript would be improved by a detailed discussion of how the THC analog derivatives affect the structural dynamics of the CB1 receptor, especially focusing on the movements and changes in the TM5 and TM6 helices.
6. In lines 318-319 of the manuscript, the discussion suggests that tight binding and slow dissociation of the ligand from the receptor are associated with hydrophobic interactions contributing to the binding free energy. If this interpretation is based on previous studies, please include appropriate references. If not, this statement should be excluded, as the current study did not perform any binding free energy calculations.
7. Please provide detailed information about the force fields used for the ligands in the all-atom Molecular Dynamics Simulations in the Materials and Methods section. This information is essential for replicating and understanding the simulations.
8. In line 127, it is stated that the shared binding site of HU210 and AM841 is presented in Extended Data Fig. 3. However, Extended Data Fig. 3 appears to show an EM density map. Please verify and correct this.
9. The statement in lines 452-453, "This resulted in a total simulation time of 11 μ s from all the simulations," may confuse readers. It would be clearer if this sentence specified that the total simulation numbers and each system was run 1.0 μ s. Please revise for clarity.

Reviewer #4

(Remarks to the Author)

In this manuscript, Dr. Gloriam and colleagues report a cryo-EM structure of the cannabinoid receptor CB1 in complex with Gi and the THC analog HU210. They further characterize the signaling properties of a series of CB1 ligands, including two endocannabinoid lipids and THC analogs, using Gi1 and β -arrestin recruitment assays. The authors elucidate critical molecular determinants underlying the potency, efficacy, and biased signaling of CB1 ligands based on the comprehensive data obtained. Additionally, they investigate ligand dissociation kinetics from CB1 for several chemically diverse ligands. This study is of high quality, offering detailed and novel insights into CB1 ligand interactions and SAR studies, which could inform the rational design of new CB1 ligands and THC analogs.

While the structural study appears robust, the pharmacological studies have some limitations. First, CB1 receptors can couple to multiple Gi/o subtypes, not just Gi1. An earlier study by Dr. Howlett's group demonstrated that different CB1 agonists may exhibit varying preferences for Gi1, Gi2, and Gi3 (doi.org/10.1124/mol.104.003558). Another study showed that only H210 induces maximal Go stimulation, with AEA and Δ 9-THC functioning only as partial agonists for the Go pathway (DOI: [10.1124/mol.56.6.1362](https://doi.org/10.1124/mol.56.6.1362)). In the CNS, Go has been suggested to be a highly abundant Gi/o subtype. Notably, another neurotransmitter GPCR, dopamine D2, has been shown to primarily couple to Go in the CNS. The authors only examined Gi1 recruitment, which may not accurately reflect CB1 signaling in the CNS. Conclusions on ligand potency and efficacy based solely on Gi1 recruitment may therefore be highly limited. Second, G protein recruitment does not necessarily correlate with the strength of cellular signaling outputs. Strong G protein recruitment may not translate to greater cAMP or Ca²⁺ responses. It would be beneficial to include additional signaling assays, such as cAMP or Ca²⁺ mobilization assays, to better characterize the efficacy of CB1 ligands.

Other issues:

1. Please include the detailed information of cryo-EM data processing in the supplementary data.
2. Cannabidiol (CBD) has been shown to function as an antagonist or inverse agonist of CB1 receptors (e.g., doi.org/10.1038/sj.bjp.0707133). However, the paper in question reports that CBD acts as an agonist. What might explain this discrepancy?

Version 2:

Reviewer comments:

Reviewer #2

(Remarks to the Author)

The authors provided excellent responses to reviewer critiques and edited the manuscript accordingly. My concerns were addressed; I recommend the resubmission be approved for publication.

Reviewer #3

(Remarks to the Author)

They addressed all of my comments, and I agree to accept the manuscript.

Reviewer #4

(Remarks to the Author)

I appreciate the additional data analysis provided by the authors. While I still believe that investigating CB1 signaling using other Gi/o subtypes and more physiologically relevant signaling assays could offer deeper and novel insights, the conclusions and findings in the revised paper, albeit limited, are well-supported.

Reviewer's Comments:

Reviewer #2 (Remarks to the Author)

The manuscript titled "Structural basis of delta 9 -THC analog activity at the Cannabinoid 1 receptor" describes the cryo-EM structure of HU210 bound to CB1R and its primary transducer Gi1. A series of docking and molecular dynamics simulations on THC and its analogs were conducted and pharmacological profiles were constructed to compare affinities, kinetics, and agonist biases.

This is not the first reported structure of a CB1R-ligand complex, but the HU210/CB1/Gi1 complex represents the largest coverage of receptor residues reported to date and moreover, this is the first disclosure of THC analog dissociation rates from an active CB1 receptor complex under experimental conditions. The methodology is sound, and there is sufficient detail provided in the methods for the work to be reproduced.

The graphical representation and data analysis is noteworthy (particularly Figure 3 and 4). The pharmacological grouping based on efficacy/potency derived from experimental data is clear.

We thank the Reviewer for the excellent summary of our paper and for noting the novel elements. We are also very pleased that the graphical artwork and pharmacological grouping are clear.

Minor Concerns:

The structure-kinetics relationship section describing AM841 to CB1R (p.6, lines 246-259) is confusing. The authors discuss the contradictory nature of the interaction between AM841 and CB1R based on several published reports (covalent vs. non-covalent) and continue this discussion (p.8, lines 319-325). If this has been settled (line 322, "This has now been disproven by AM841-CB1 structure complexes"), then why spend time in a discussion? If this has not been settled, why haven't the authors considered reversible covalent inhibition (which is observed routinely with isothiocyanates). If this compound is a point of contention, why include it as one of the 10 analogs?

AM841 was included to provide additional evidence of its reversible binding. However, it is correct that this has already been proven in the past decade from both structural and pharmacological data. For that reason, we have now removed the first paragraph of the structure-kinetics relationship section, except the last sentence, which introduces the experiments made herein.

I am not convinced that the statement "This may advance our understanding of fundamental receptor function and support the further development of THC and its analogs as therapeutics." is necessary.

We think that some concluding sentence of the Introduction is beneficial to explain the impact and relevance to readers that do not have a background in structure-activity relationships. However, we have toned down and clarified this sentence which now reads: "*These structure-activity relationships serve to guide further development of THC analogs as tool compounds and therapeutics.*"

It is important to specify delta9 -THC and analogs vs. delta8 -THC and analogs. Although delta8 -THC is indeed an analog of delta9 -THC, specific designations are helpful to avoid ambiguity. For example, Fig. 2 and Fig. 3 (delta-9 THC analogs) and the Extended Data Fig. 1 (p. 22) is titled "Chemical structures of delta9-THC analogs and additional agonists assayed at CB1". There are a number of delta8 -THC analogs illustrated in these figures (including the main compound studied, HU210). Furthermore, it may be beneficial to edit the title to reflect the concerns listed above.

We thank Reviewer #2 for pointing this out. To resolve the ambiguity, we have edited both figures and parts of the manuscript text to distinguish between delta9 and delta8 compounds. The manuscript title has been edited to just "THC analogs" – removing the mention of delta9.

Reviewer #3 (Remarks to the Author)

I have reviewed the manuscript by Thor S. Thorsen et al., which investigates the cryoEM structure of the Gi1-bound CB1 receptor with HU210. The study provides a comprehensive pharmacological profiling of this receptor, comparing its efficacy and potency with 12 known analogues, including 10 exogenous and 2 endogenous ligands. The authors employ BRET techniques to compare biased signaling and kinetics and utilize all-atom molecular dynamics simulations over 1000 ns to elucidate the structure-activity relationship of the exogenous analogues. Then atomic level pharmacological explanations described based on the contact frequency between binding pocket residues and ligands.

This work represents a valuable contribution to the understanding of GPCR structure and pharmacology. We thank Reviewer #3 for outlining the different, interdisciplinary components of our study and for confirming its value to the understanding of GPCR structure and pharmacology.

However, several concerns need to be addressed before the manuscript can be considered for publication. You can find comments below:

1. Although the authors conducted 1000 ns all-atom molecular dynamics simulations, they did not present detailed data or results from these simulations. First, the methods section should specify the number of replicates for each system (each drug) of MD simulation.

In our first submission, we had run one replicate for each ligand. In this revision, we have tripled the simulations performed to attain triplicates of each ligand system and we note this in Methods. Detailed data and results from these simulations have also been added, as answered under the relevant comments below.

2. The reproducibility of the system should be demonstrated, ideally in the supplementary materials, using RMSD data for CB1/Gi1 with drug complexes throughout the simulation time. Given the complexity of GPCR membrane systems and the relatively less-sampled conformational dynamics, this potentially be affect to results of interactions analysis between comparisons of structurally similar analogous compounds and binding pocket residues

We have added graphs of RMSD data for CB1/Gi1 with drug complexes throughout the simulation time to Extended Data Fig. 9.

3. The description of the spatiotemporal interactions between the THC analog and the CB1 receptor, as shown in Fig. 4c, is insufficient. The authors should include data on contact frequency throughout the simulation time to provide a more comprehensive understanding and confidence.

We have added graphs to Extended Data Fig. 10 showing time-resolved frequencies for all 29 ligand-residue contacts in Fig. 4b over the course of 1000 ns and across all three replicates for THC and its analogs. Furthermore, we are currently in the process of uploading the complete trajectories for all 11 simulated ligands in GPCRmd.

4. It would be beneficial if the authors could provide time series data for hydrogen bond analysis, particularly for Fig. 5(b), 5(f), and 5(g).

Graphs of time series data for the hydrogen bonds in Fig. 5(b), 5(c-e), 5(f), and 5(g) have been added to the Extended Data Fig. 10, which has five panels, as Fig. 5(f) contains two ligands.

5. The manuscript would be improved by a detailed discussion of how the THC analog derivatives affect the structural dynamics of the CB1 receptor, especially focusing on the movements and changes in the TM5 and TM6 helices.

We agree that this would be interesting. However, studying whole-helix movements would require a considerably longer timescale beyond our computing capability for the large system numbers and sizes. For this revision, we attempted to correlate E_{\max} and pEC_{50} values with contact frequencies of the residue pairs $L^{6x37}-R^{3x50}$, $Y^{5x58}-R^{3x50}$ and $W^{6x48}-V^{3x40}$ that constitute activation “microswitches” <https://doi.org/10.1038/>

s41594-021-00674-7. Unfortunately, this did not result in correlations. This indicates that other approaches would be necessary to correlate ligand interactions with helix movements. Furthermore, the mechanisms of receptor activation and signal transduction onto the G protein is outside of the scope of this study, which focuses on ligand-receptor structure-activity relationships. We therefore think that this would need to be answered in a separate approach and study.

6. In lines 318-319 of the manuscript, the discussion suggests that tight binding and slow dissociation of the ligand from the receptor are associated with hydrophobic interactions contributing to the binding free energy. If this interpretation is based on previous studies, please include appropriate references. If not, this statement should be excluded, as the current study did not perform any binding free energy calculations. This was not based on previous studies. We simply mean that hydrophobic ligand moieties that escape solvent and pack in a hydrophobic binding pocket contribute are likely to contribute to affinity and slow kinetics. However, we have now removed the part “*contributing to the free energy of binding*”.

7. Please provide detailed information about the force fields used for the ligands in the all-atom Molecular Dynamics Simulations in the Materials and Methods section. This information is essential for replicating and understanding the simulations.

The force field, OPLS3e was already mentioned in the Methods along with a reference. Schrödinger does not provide access to viewing OPLS3 and newer force field parameters that are used by the Desmond molecular dynamics program within the suite. This is mentioned on Schrödinger’s support page: <https://support.schrodinger.com/s/article/809>. The force field parameters are encoded in the simulation input files which are not human readable. As a result, unfortunately in this case, we cannot retrieve the parameters and present them in supplementary tables.

8. In line 127, it is stated that the shared binding site of HU210 and AM841 is presented in Extended Data Fig. 3. However, Extended Data Fig. 3 appears to show an EM density map. Please verify and correct this. We had referenced the EM density map in Extended Data Fig. 3b because three binding site residues that make contacts with HU210 also does this with AM841. However, we have now removed the figure reference and added a reference to the publication of the AM841-bound CB₁ structure (10.1016/j.cell.2020.01.008).

9. The statement in lines 452-453, “This resulted in a total simulation time of 11 μ s from all the simulations,” may confuse readers. It would be clearer if this sentence specified that the total simulation numbers and each system was run 1.0 μ s. Please revise for clarity.

We have clarified this (and added updated based on the three MD replicas) to: “*This resulted in a total sampling of 3 μ s for each of the 11 receptor-ligand systems.*”

Reviewer #4 (Remarks to the Author):

In this manuscript, Dr. Gloriam and colleagues report a cryo-EM structure of the cannabinoid receptor CB₁ in complex with Gi and the THC analog HU210. They further characterize the signaling properties of a series of CB₁ ligands, including two endocannabinoid lipids and THC analogs, using Gi1 and β -arrestin recruitment assays. The authors elucidate critical molecular determinants underlying the potency, efficacy, and biased signaling of CB₁ ligands based on the comprehensive data obtained. Additionally, they investigate ligand dissociation kinetics from CB₁ for several chemically diverse ligands. This study is of high quality, offering detailed and novel insights into CB₁ ligand interactions and SAR studies, which could inform the rational design of new CB₁ ligands and THC analogs.

We thank Reviewer #4 for describing especially the experimental aspects of our work, and for stressing the high quality and novel insights and their potential in guiding ligand design.

While the structural study appears robust, the pharmacological studies have some limitations. First, CB1 receptors can couple to multiple Gi/o subtypes, not just Gi1. An earlier study by Dr. Howlett's group demonstrated that different CB1 agonists may exhibit varying preferences for Gi1, Gi2, and Gi3 (doi.org/10.1124/mol.104.003558). Another study showed that only HU210 induces maximal Go stimulation, with AEA and Δ9-THC functioning only as partial agonists for the Go pathway (DOI: [10.1124/mol.56.6.1362](https://doi.org/10.1124/mol.56.6.1362)). In the CNS, Go has been suggested to be a highly abundant Gi/o subtype. Notably, another neurotransmitter GPCR, dopamine D2, has been shown to primarily couple to Go in the CNS. The authors only examined Gi1 recruitment, which may not accurately reflect CB1 signaling in the CNS. Conclusions on ligand potency and efficacy based solely on Gi1 recruitment may therefore be highly limited.

We have added a new analysis of whether the Gi1-based SAR also reflects Go coupling, which may be more prevalent in the CNS, in a new section "*Gi1 recruitment correlates with Go recruitment and cAMP inhibition*" and the associated Extended Data Figure 7. EDF7 shows scatter plots of E_{max} and pEC₅₀ values for Gi1 and Go recruitment, respectively. The Gi1 and Go_A data come from the current and our previous study (DOI: [10.1101/2020.11.09.375162](https://doi.org/10.1101/2020.11.09.375162)), respectively and the same BRET-based G protein-recruitment assay. They range the endocannabinoids 2-AG and Anandamide, WIN55212-2, THC and THC analogs Cannabinol, CP55940, Nabilone and HU210. We find that Gi1 and Go correlate well with R² values of 0.72 and 0.89 for E_{max} and pEC₅₀, respectively. This corroborates that the SAR described in this study for Gi1 also applies to Go.

The remainder of the manuscript and figure set remain focused on Gi1 for two reasons. Firstly, Gi is more often than Go used to represent the Gi/o family in literature making the results presented herein more comparable to other studies. Secondly, Gi is better represented among structure complexes with GPCRs than Go. There are four Gi1 and no Go structures in complex with CB1. This includes the structure determined herein of the HU210-CB1-Gi1 complex. Furthermore, there are one Gi2 and no Gi3 structure complexes with CB1.

Second, G protein recruitment does not necessarily correlate with the strength of cellular signaling outputs. Strong G protein recruitment may not translate to greater cAMP or Ca²⁺ responses. It would be beneficial to include additional signaling assays, such as cAMP or Ca²⁺ mobilization assays, to better characterize the efficacy of CB1 ligands.

We have added a new analysis examining if G protein recruitment correlates with the strength of cellular signaling outputs in the second paragraph of the new section "*Gi1 recruitment correlates with Go recruitment and cAMP inhibition*" and the associated Extended Data Figure 8. EDF8 shows scatter plots of E_{max} and pEC₅₀ values for Gi1 recruitment and cAMP inhibition, respectively. The Gi recruitment data stems from the current study, whereas cAMP inhibition data is derived from literature ([doi:10.1124/mol.115.099192](https://doi.org/10.1124/mol.115.099192) and [doi:10.1111/bph.15066](https://doi.org/10.1111/bph.15066)). These plots include the endocannabinoids 2AG and Anandamide, WIN55212-2, THC, HU210, and CP55940. We find that Gi recruitment and cAMP inhibition correlate well with both E_{max} (r² = 0.99 and 0.85) and pEC₅₀ (r² = 0.88) values.

Other issues:

1. Please include the detailed information of cryo-EM data processing in the supplementary data.

We have added detailed information for the cryo-EM data processing in Extended Data Figure 6.

2. Cannabidiol (CBD) has been shown to function as an antagonist or inverse agonist of CB1 receptors (e.g., doi.org/10.1038/sj.bjp.0707133). However, the paper in question reports that CBD acts as an agonist. What might explain this discrepancy?

This is a misunderstanding, probably due to the similar naming of Cannabidiol and Cannabinol. We have only tested Cannabinol, which is an agonist of CB₁. We would also like to note that a recent publication has found that CBD acts as a negative allosteric modulator (DOI:10.1111/bph.13250).